# Achromatic and Athermal Design of Aerial Catadioptric Optical Systems by Efficient Optimization of Materials

**DOI:** 10.3390/s23041754

**Published:** 2023-02-04

**Authors:** Jing Li, Yalin Ding, Xueji Liu, Guoqin Yuan, Yiming Cai

**Affiliations:** 1Changchun Institute of Optics, Fine Mechanics and Physics, Chinese Academy of Sciences, Jilin 130033, China; 2Key Laboratory of Airborne Optical Imaging and Measurement, Chinese Academy of Sciences, Jilin 130033, China; 3University of Chinese Academy of Sciences, Beijing 100049, China; 4Key Laboratory of Space-Based Integrated Information System, Institute of Software, Chinese Academy of Sciences, Beijing 100190, China

**Keywords:** temperature change, catadioptric optical system, achromatic, athermal design, aerial camera

## Abstract

The remote sensing imaging requirements of aerial cameras require their optical system to have wide temperature adaptability. Based on the optical passive athermal technology, the expression of thermal power offset of a single lens in the catadioptric optical system is first derived, and then a mathematical model for efficient optimization of materials is established; finally, the mechanical material combination (mirror and housing material) is optimized according to the comprehensive weight of offset with temperature change and the position change of the equivalent single lens, and achieve optimization of the lens material on an athermal map. In order to verify the effectiveness of the method, an example of a catadioptric aerial optical system with a focal length of 350 mm is designed. The results show that in the temperature range of −40 °C to 60 °C, the diffraction-limited MTF of the designed optical system is 0.59 (at 68 lp/mm), the MTF of each field of view is greater than 0.39, and the thermal defocus is less than 0.004 mm, which is within one time of the focal depth, indicating that the imaging quality of the optical system basically does not change with temperature, meeting the stringent application requirements of the aerial camera.

## 1. Introduction

Aerial cameras play an irreplaceable role in many fields, such as environmental protection, ecological resource investigation, land and resources management, geological disaster monitoring, climate impact, and GIS construction by acquiring geographic remote sensing information [1,2,3,4,5]. The main payload of the aerial camera is the remote sensing camera with wide temperature adaptability, which widely uses the catadioptric optical system, and has outstanding advantages in reducing the volume size and correcting the aberration, etc. [6,7,8,9,10]. Compared with the ground optical system, the ambient temperature of the camera changes more widely due to the continuous change of the flight altitude of the carrier. At the same time, the use of the aerial camera needs to be suitable for different environmental and climatic conditions. Its harsh use conditions cause the degradation of the imaging quality of the optical system. Therefore, it is necessary to reduce the influence of temperature on aerial optical systems so that aerial cameras can be applied to a wider temperature range [11,12,13].

In order to reduce the influence of temperature on optical system, scholars have carried out some research on athermal design methods for cameras. The existing athermal design methods are mainly divided into three categories: mechanical passive athermal technology, electromechanical active athermal technology, and optical passive athermal technology. Compared with the other two thermal technologies, the optical passive athermal technology has the advantages of simple structure, low cost, stability, and reliability, so the optical passive athermal technology is usually adopted as the optimization method in the airborne optical system [14,15,16].

Optical passive athermal technology is based on the change in the refractive index of lens materials and the expansion or contraction of optical elements and housing materials [17,18,19]. Considering the materials of the mirror and the housing, Qiuzhen Xu analyzed the thermal difference of the classical Cassegrain system and pointed out that when the thermal expansion coefficients of the mirror and the housing material are the same, the thermal difference is always zero [20]. On this basis, the athermal design of a catadioptric optical system is usually divided into two parts. First, match the mirror material with the housing material between the primary and secondary mirrors to make their thermal expansion coefficients as equal as possible. Secondly, the athermal design is carried out for the lens part to match the temperature effect of the lens with the thermal expansion coefficient of the lens housing material [21,22]. On the basis of separate considerations of athermalization, the housing of a catadioptric optical system is mostly composed of two or more materials, which affects the structural stability of the system. When selecting a material for the housing, the selection of the mirror material needs to consider the influence of the temperature of the lens in the optical system. Therefore, how to efficiently optimize the mirror and housing materials according to the actual situation is an important research topic for people designing athermal catadioptric optical systems.

The research on optimizing lens materials is mostly based on the athermal map, which usually requires that the housing materials and optical materials meet specific relationships in the graph where the abscissa is chromatic power and the ordinate is thermal power. For example, Tamagawa pointed out that the optical power distribution of the three lens system should make the optical power of each lens the smallest, and the three materials that form the largest triangle area should be selected on an athermal map, which is not applicable to multilens systems [23,24]; Lim et al. proposed the linear relationship between the housing material and the optical material based on the equivalent of the system structure [25,26]. On this basis, Na Xie proposed a structure grouping method based on the weights of thermal and chromatic power [27]. Yang Zhu proposed a structure grouping method based on combined glasses and comprehensive distance weight [28]. The above athermal method based on the athermal map is conducive to improving the efficiency of lens material optimization. However, it is not suitable for catadioptric optical systems. When the mirror exists, the position of the ideal lens material on an athermal map will shift linearly, which will affect the effect of the athermal design.

Automatic optimization can be achieved in optical design software by creating multiple structures at different temperatures [29,30,31]. Relying only on software to optimize usually takes a long time and requires high optimization algorithms. Therefore, optical designers usually manually replace materials according to the athermal equation and then further optimize in software [32]. In order to avoid large changes in system parameters, homologous materials are given priority in material selection, but when the thermal power of homologous materials changes greatly, the optical system cannot achieve the optimal athermal effect.

To sum up, the existing athermal design methods cannot be effectively applied to catadioptric optical systems. To address these issues, this paper proposes an efficient optimization of materials to achieve the achromatic and athermal design of catadioptric optical systems. This method establishes the mathematical model of the athermal design. In order to select the optimal mirror and housing material more quickly, a combined analysis of the mirror and housing material is proposed. The athermal ability of the combined material is evaluated based on the comprehensive weight of the change of the thermal power offset of the single lens with temperature and change in position of the equivalent single lens. The position of the ideal lens material for a catadioptric optical system is determined according to the expression of the thermal power offset of the single lens; the material optimization is realized on an athermal map, and the effectiveness of the method is proved by simulation experiments and design examples.

## 2. Achromatic and Athermal Theory

This section mainly introduces the process of achromatic and athermal design of the catadioptric system proposed in this paper, gives the conditions of achromatic and athermal, and reveals the key role of the thermal power offset of the single lens in catadioptric optical systems by constructing equivalent singlet and athermal glass map.

### 2.1. Basic Achromatic and Athermal Process of Catadioptric Systems

The traditional optical design usually optimizes the optical system at room temperature. In order to adapt to a wider temperature range, athermalization and optimization must be performed. Figure 1 describes the detailed process of the athermal design method for the catadioptric optical system proposed in this paper, which can more clearly represent the athermal design idea of this paper. The specific steps are as follows, and the details of each step will be described in subsequent chapters of this paper.

Establish mathematical models for the achromatic and athermal design of catadioptric optical systems, as shown in Section 2.2, Section 2.3 and Section 3.1.Establish evaluation method for material optimization of catadioptric optical systems, as shown in Section 3.2.Build an athermal distribution map of a glass catalog in the visible light band, as shown in Section 4.2.Evaluate the thermal properties of mirror material and housing material and determine the optimal combination of mirror and housing materials, as shown in Section 4.3.Evaluate the effect of weighted optical power on the position of the equivalent singlet lens, and perform lens material optimization and power redistribution, as shown in Section 4.4.Assess image quality and temperature adaptability of optical systems, as shown in Section 4.5.

### 2.2. Achromatic and Athermal Conditions

For a thin lens with optical power $\phi_{i}$ and Abbe number $\nu_{i}$, the chromatic power $\omega_{i}$ and thermal power $\gamma_{i}$ are given as follows [23,24,25]:(1)$$\begin{array}{l} \omega_{i}=\frac{1}{\nu_{i}}\\ \gamma_{i}=\frac{\left(\frac{\partial \phi_{i}}{\partial T}\right)}{\phi_{i}}=\frac{\frac{\partial n_{i}}{\partial T}}{n_{i}-1}-\alpha_{i} \end{array}$$
where $n_{i}$ is the refractive index at the center wavelength, $\alpha_{i}$ is the coefficient of thermal expansion (CTE) of the lens material, and T is the temperature.

Within a certain wavelength range, the Abbe number of the material and the refractive index of the central wavelength can be inquired by the glass manufacturer. The thermal power of the mirror is related to its own material and is equal to the thermal expansion coefficient of the material. There is no chromatic aberration in the reflection system, and its chromatic power is always 0. Therefore, for a catadioptric optical system with *K* lenses, the total optical power of $\phi_{T}$, the chromatic power of a mirror is 0, and the thermal power is α, the equation of the total optical power and the achromatic and athermal conditions are given as follows:(2)$$\left\{\begin{array}{c} \phi_{T}=\sum_{i=1}^{k}\phi_{i}^{\prime }+\varphi_{1}^{\prime }+\varphi_{2}^{\prime }\\ \frac{d\phi_{T}}{d\lambda }=\sum_{i=1}^{k}\omega_{i}\phi_{i}^{\prime }=0\\ \frac{d\phi_{T}}{dT}=\sum_{i=1}^{k}\gamma_{i}\phi_{i}^{\prime }-\alpha \left(\varphi_{1}^{\prime }+\varphi_{2}^{\prime }\right)=-\alpha_{m}\phi_{T} \end{array}\right.$$
where $\phi_{i}^{\prime }=\left(\frac{h_{i}}{h_{1}}\right)\phi_{i}$, $\varphi_{1}^{\prime }=\left(\frac{h_{PM}}{h_{1}}\right)\varphi_{1}$, and $\varphi_{2}^{\prime }=\left(\frac{h_{SM}}{h_{1}}\right)\varphi_{2}$ are the weighted optical power of the lens, the primary mirror, and the secondary mirror, respectively. $h_{i}$ is the paraxial ray height at the *i*’th lens, $h_{PM}$ is the paraxial ray height at the primary mirror, and $h_{SM}$ is the paraxial ray height at the secondary mirror. α is the CTE of the mirror material, and $\alpha_{m}$ is the CTE of the housing material.

### 2.3. Equivalent Single Lens and Athermal Glass Map

This paper describes the achromatic and athermal conditions of a catadioptric optical system based on an equivalent single lens. Since the reflection system has no chromatic aberration, the achromatic equation does not need to consider the influence of the reflection system, and the athermal equation needs to consider the influence of reflection, refraction, and housing. The weighted optical power, chromatic power, and thermal power of a single lens $L_{j}$ are $\phi_{j}^{\prime }$, $\omega_{j}$, and $\gamma_{j}$ respectively, and the remaining K−1 lenses are equivalent to a single lens $L_{e}$. The weighted optical power $\phi_{e}^{\prime }$, equivalent chromatic power $\omega_{e}$ , and equivalent thermal power $\gamma_{e}$ of the equivalent single lens $L_{e}$ are given as follows [24,25,26]:(3)$$\phi_{e}^{\prime }=\sum_{i=1}^{k}\phi_{i}^{\prime }-\phi_{j}^{\prime }$$
(4)$$\omega_{e}=\frac{\sum_{i=1}^{k}\omega_{i}\phi_{i}^{\prime }-\omega_{j}\phi_{j}^{\prime }}{\phi_{e}^{\prime }}$$
(5)$$\gamma_{e}=\frac{\sum_{i=1}^{k}\gamma_{i}\phi_{i}^{\prime }-\gamma_{j}\phi_{j}^{\prime }}{\phi_{e}^{\prime }}$$

Therefore, a catadioptric optical system with K lenses and total optical power $\phi_{T}$ can be recomposed of the single lens $L_{j}$, the equivalent single lens $L_{e}$ , and the reflection system. Equation (2) can be rewritten as follows:(6)$$\phi_{T}=\phi_{j}^{\prime }+\phi_{e}^{\prime }+\left(\varphi_{1}^{\prime }+\varphi_{2}^{\prime }\right)$$
(7)$$\frac{d\phi_{T}}{d\lambda }=\omega_{j}\phi_{j}^{\prime }+\omega_{e}\phi_{e}^{\prime }=0$$
(8)$$\frac{d\phi_{T}}{dT}=\gamma_{j}\phi_{j}^{\prime }+\gamma_{e}\phi_{e}^{\prime }-\alpha \left(\varphi_{1}^{\prime }+\varphi_{2}^{\prime }\right)=-\alpha_{m}\phi_{T}$$

From Equation (7), the relationship between the chromatic power and the weighted power of the singlet and the equivalent singlet is as follows:(9)$$\frac{\phi_{e}^{\prime }}{\phi_{j}^{\prime }}=-\frac{\omega_{j}}{\omega_{e}}$$

Substituting Equations (6) and (9) into Equation (8), the expression for $\gamma_{j}$ is as follows:(10)$$\gamma_{j}=\frac{\gamma_{e}+\alpha_{m}}{\omega_{e}}\omega_{j}-\alpha_{m}+\frac{\left(\alpha -\alpha_{m}\right)\left(\varphi_{1}^{\prime }+\varphi_{2}^{\prime }\right)}{\phi_{j}^{\prime }}$$

The expression for $\gamma_{j}$ in a refractive optical system is as follows [25,27]:(11)$$\gamma_{j}=\frac{\gamma_{e}+\alpha_{m}}{\omega_{e}}\omega_{j}-\alpha_{m}$$

It can be seen from Equations (10) and (11) that compared with the refractive system, the thermal power of the single lens in the catadioptric optical system has an offset $\Delta =\frac{\left(\alpha -\alpha_{m}\right)\left(\varphi_{1}^{\prime }+\varphi_{2}^{\prime }\right)}{\phi_{j}^{\prime }}$, which is related to the weighted optical power of the single lens, the primary and secondary mirrors, and the CTE of the mirror and housing materials.

According to Equations (10) and (11), the achromatic and athermal conditions of the catadioptric and refractive optical system on the athermal map are drawn in Figure 2, the abscissa is the chromatic power, and the ordinate is the thermal power. $L_{j}\left(\omega_{j},\gamma_{j}\right)$, $L_{e}\left(\omega_{e},\gamma_{e}\right)$, and $H\left(0,-\alpha_{m}\right)$, respectively, represent the coordinates of a single lens $L_{j}\left(\omega_{j},\gamma_{j}\right)$, equivalent single lens $L_{e}\left(\omega_{e},\gamma_{e}\right)$, and housing $H\left(0,-\alpha_{m}\right)$ on the athermal map.

It can be seen from Figure 2 that $L_{j}\left(\omega_{j},\gamma_{j}\right)$, $L_{e}\left(\omega_{e},\gamma_{e}\right)$, and $H\left(0,-\alpha_{m}\right)$ are located on a line in the refractive optical system, $L_{j}\left(\omega_{j},\gamma_{j}\right)$, $L_{e}\left(\omega_{e},\gamma_{e}\right)$, and $H\left(0,-\alpha_{m}\right)$ are not located on a line in the catadioptric optical system. $L_{j}\left(\omega_{j},\gamma_{j}\right)$ lies on the line determined by $L_{e}\left(\omega_{e},\gamma_{e}\right)$ and $H\left(0,-\alpha_{m}\right)$ offset by $\Delta =\frac{\left(\alpha -\alpha_{m}\right)\left(\varphi_{1}^{\prime }+\varphi_{2}^{\prime }\right)}{\phi_{j}^{\prime }}$ along the ordinate.

On the one hand, the expression of the thermal power offset of the single lens points out that the conditions for the achromatic and athermal design of catadioptric optical systems and refractive optical systems are different, and on the other hand, it extends the graphical athermal design method, which is a prerequisite for realizing achromatic and athermal design of catadioptric optical systems based on efficient optimization of materials.

## 3. Achromatic and Athermal Design of Catadioptric Optical Systems by Efficient Optimization of Materials

The offset described in the previous section varies with temperature, resulting in a change in the position of the equivalent single lens that satisfies the achromatic and athermal conditions, which is unfavorable for realizing the wide-range temperature adaptability of the system. Yang Zhu proposed an athermalization and achromatization method of a refractive optical system based on combined glasses and comprehensive distance weight to select and replace optical and housing materials quantitatively [28]. On this basis, this section introduces the mathematical model of the achromatic and athermal design of catadioptric optical systems, which reveals the boundary conditions for material optimization, and then introduces the evaluation method for material optimization, which is used to determine the athermal ability of materials.

### 3.1. Mathematical Model for Achromatic and Athermal Design of a Catadioptric Optical System

A new graphical method is proposed to perform material optimization within a reasonable range of Δ, while keeping the total optical power unchanged. When the offset varies with temperature, the achromatic and athermal conditions on an athermal glass map are shown in Figure 3.

It can be seen from Figure 3, $L_{1}\left(\omega_{j},\gamma_{j}-\Delta \right)$, $L_{e}\left(\omega_{e},\gamma_{e}\right)$, $H\left(0,-\alpha_{m}\right)$ are located on a line, the thermal power $\gamma_{e}$ of the equivalent single lens can be expressed as follows:(12)$$\gamma_{e}=\frac{\gamma_{e}-\gamma_{j}+\Delta }{\omega_{e}-\omega_{j}}\omega_{e}-\alpha_{m}$$

Solve for $\alpha_{m}$ to obtain Equation (13):(13)$$\alpha_{m}=\frac{\gamma_{e}\omega_{j}-\gamma_{j}\omega_{e}+\Delta \omega_{e}}{\omega_{e}-\omega_{j}}$$

Δ is related to the weighted optical power of the single lens, the primary and secondary mirrors, and the CTE of the mirror and housing materials. When the mirror material and the housing material are fixed, the weighted optical power changes with temperature, which causes the position of the equivalent single lens of the system after the temperature change to be inconsistent with the system before the temperature change. When the achromatic and athermal conditions are not met, ensure that the variation of Δ with temperature is the smallest and the position of the equivalent single lens before and after the temperature change is as close as possible to make the system adapt to a wider temperature range, and the position adjustment of the equivalent single lens can be realized by the material optimization. Therefore, the realization of an achromatic and athermal design should satisfy the following constraints:

(1) The total optical power remains unchanged:(14)$$\phi_{T}=\sum_{i=1}^{k}\phi_{i}^{\prime }+\varphi_{1}^{\prime }+\varphi_{2}^{\prime }=\mathrm{constant}$$

(2) The position change of the equivalent single lens is minimal:(15)$$L_{e}-L_{e,origin}=\sqrt{{\left(\omega_{e}-\omega_{e,origin}\right)}^{2}+{\left(\gamma_{e}-\gamma_{e,origin}\right)}^{2}}=\min$$

(3) The change of offset with temperature is the smallest:(16)$$L_{e}^{\prime }-L_{e}=\sqrt{{\left(\omega_{e}^{\prime }-\omega_{e}\right)}^{2}+{\left(\gamma_{e}^{\prime }-\gamma_{e}\right)}^{2}}=\min$$

(4) The position of the equivalent single lens changes minimally with temperature:(17)$$L_{e}^{\prime }-L_{e}=\sqrt{{\left(\omega_{e}^{\prime }-\omega_{e}\right)}^{2}+{\left(\gamma_{e}^{\prime }-\gamma_{e}\right)}^{2}}=\min$$

(5) The CTE of the housing material satisfies Equation (18):(18)$$\alpha_{m}=\frac{\left(\sum_{i=1}^{k}\gamma_{i}\phi_{i}^{\prime }-\gamma_{j}\phi_{j}^{\prime }\right)\omega_{j}-\left(\gamma_{j}-\Delta \right)\left(\sum_{i=1}^{k}\omega_{i}\phi_{i}^{\prime }-\omega_{j}\phi_{j}^{\prime }\right)}{\sum_{i=1}^{k}\omega_{i}\phi_{i}^{\prime }-\omega_{j}\phi_{j}^{\prime }-\omega_{j}\phi_{e}^{\prime }}$$

The optimization of the mirror, housing materials, and the equivalent single lens should satisfy the above equation simultaneously. Different from previous studies, the criterion for material optimization proposed by the method in this paper is not a fixed value, but a minimum value, so it can be used to evaluate the athermal ability of the material, that is, the smaller the Equations (15)–(17) after material optimization, the better the athermal effect of the system is. Through the evaluation of athermal ability, sequential simulation of materials is avoided, and the efficiency of material optimization is improved.

### 3.2. Evaluation Method for Material Optimization

#### 3.2.1. Evaluation Method of Mirror and Housing Material Combinations

It can be seen from the expressions of the offset Δ and the change $\frac{d\Delta }{dT}$ of the offset with the temperature that when the optical system is fixed, the CTE of the mirror material and the housing material will affect the temperature adaptability of the system. Therefore, before changing the glass material, it is necessary to evaluate the thermal ability of the mirror material and the housing material to determine the best combination of the mirror and the housing materials. Since the position of the equivalent single lens of the catadioptric system will change with temperature, in order to adapt to a wider temperature range, the change in the position of the equivalent single lens needs to be as small as possible. At the same time, when there is a deviation between the position of the initial equivalent singlet and the position of the ideal equivalent singlet, the smaller the deviation, the smaller the adjustment amount required for the athermalization of the system. Therefore, the evaluation and selection of mirror and housing materials can be quantified from three aspects, as shown in Figure 4: The change in offset with temperature $d_{1}=\frac{d\Delta }{dT}$, the distance $d_{2}$ from the equivalent singlet $L_{e,origin}(\omega_{e,origin},\gamma_{e,origin})$ to the ideal equivalent singlet $L_{e}(\omega_{e},\gamma_{e})$, and the distance $d_{3}$ of the ideal equivalent singlet $L_{e}(\omega_{e},\gamma_{e})$ to the equivalent single lens $L_{e}^{\prime }(\omega_{e}^{\prime },\gamma_{e}^{\prime })$ after a temperature change.

The specific steps of the selection method of mirror and housing material combinations are as follows:

(1) Determine the number of optical system groupings. Assuming that the optical system includes k lenses, the number of combinations of equivalent single lenses is $C_{k}^{k-1}=\frac{\left(k\right)!}{(k-1)!}=k$, so $d_{1}$, $d_{2}$ and $d_{3}$ can be divided into k groups.

(2) Define distance ratios. When there are m commonly used mirror materials and n commonly used housing materials, there are a total of m×n combinations of mirror and housing materials. For m×n×k kinds of distance results, the calculated distance ratios are, respectively:(19)$$\begin{array}{l} Rate1_{i}=1-\frac{d1_{i}}{\max(d1_{i})}\\ Rate2_{i}=1-\frac{d2_{i}}{\max(d2_{i})}\\ Rate3_{i}=1-\frac{d3_{i}}{\max(d3_{i})} \end{array}$$

(3) Define comprehensive weight. Add the distance ratios of the k groups to obtain the comprehensive weights of m×n combinations of mirror and housing materials:(20)$$Weight_{i}=SUM\left(Rate1_{\left(\mathrm{group}\;i\right)}+Rate2_{\left(\mathrm{group}\;i\right)}+Rate3_{\left(\mathrm{group}\;i\right)}\right)$$

From the definition of distance ratios and comprehensive weight, it can be known that when the comprehensive weight is larger, the athermal ability of the material combination is better. The calculation of the comprehensive weights provides an evaluation method for the optimization of mirror and housing materials.

#### 3.2.2. Evaluation Method of Lens Materials

The evaluation method of lens material relies on an athermal map. When the mirror and housing materials are determined, it is necessary to select the grouping of the optical system with the largest comprehensive weight, determine the position of the single lens and the equivalent single lens, and realize the optimization of lens material on an athermal map. The evaluation method for lens materials is shown in Figure 5.

When mirror and housing materials and optical system groupings are determined, the key positions $H\left(0,-\alpha_{m}\right)$, $L_{j}\left(\omega_{j},\gamma_{j}\right)$, $L_{e}(\omega_{e},\gamma_{e})$, and $L_{e}^{\prime }(\omega_{e}^{\prime },\gamma_{e}^{\prime })$ are determined, so the evaluation method for lens material optimization is $d_{2}\approx 0$, that is, $L_{e,origin}(\omega_{e,origin},\gamma_{e,origin})$ and $L_{e}(\omega_{e},\gamma_{e})$ coincide.

This section solves the problem of the wide-range temperature adaptability of the catadioptric system by establishing the mathematical model of achromatic and athermal design and the evaluation method for material optimization, which is the theoretical basis for realizing athermal design of the catadioptric optical system by efficient optimization of materials.

## 4. Design Examples and Analysis

In order to verify the effectiveness of the efficient material optimization method to achieve athermal design, this section presents an example of the athermal design of an aerial optical system based on this method. Firstly, the initial catadioptric optical system and visible light athermal glass map are established, and then the mirror, housing material, and lens material are efficiently optimized. Finally, the design results are given, which proves the effectiveness of the athermal design method in this paper.

### 4.1. Optical Specifications and Initial Design

The layout of the initial aerial camera optical system is shown in Figure 6, including the primary and secondary mirrors and four lenses, as an athermal example working in a large temperature range from −40 °C to 60 °C. The focal length was 350 mm, the F-number was 4, the full field of view was 6°, and the wavelength range of 550–750 nm. The initial housing material is titanium alloy with CTE $\alpha_{m}=9.1\times 10^{-6}/{}^{\circ}C$, and the mirror material is fused silica with CTE $\alpha =0.58\times 10^{-6}/{}^{\circ}C$. All lens surfaces are spherical, the primary and secondary mirrors are quadratic surfaces, and the conic coefficients are −4.464 and −277.283, respectively. The total length is 380.121 mm.

Table 1 summarizes the optical properties of the elements, including the glass material, chromatic power ω, thermal power γ, optical power ϕ, and paraxial ray height *h*.

Figure 7 depicts the modulation transfer function (MTF) of the initial optical system at 68 lp/mm when the temperature is 20 °C, −40 °C, and 60 °C, respectively. The initial optical system was optimized at 20 °C, the same as the environmental temperature in the room, and the MTFs of all fields were greater than 0.4 at 68 lp/mm. However, at −40 °C and 60 °C, the MTF decreased to around 0, as shown in Figure 7b,c. It can be seen that the imaging quality is greatly affected by the environmental temperature without the athermal design or optimization of the optical system.

### 4.2. Establishment and Analysis of Visible Glass Catalog

In order to improve the success rate of athermal designs, there should be enough glass materials to choose from the athermal map. Therefore, this paper selects the glass material catalog of Chengdu Guangming in 2021, as shown in Figure 8a [33], with a total of 240. Compared with the 103 materials of Schott [34], Chengdu Guangming’s glass warehouse has more choices. Figure 8b is n_d_/v_d_ map of Chengdu Guangming [33]. It can be obtained through the official website and imported into zemax for subsequent optimization design.

According to the optical properties of the material itself, the thermal power and chromatic power are calculated by Equation (1), and the athermal glass map of the Chengdu Guangming catalog is shown in Figure 9.

It can be seen from Figure 9 that the thermal power of the 240 materials ranges from −40 to 5, and the chromatic power range from 5 to 60. The thermal power of most kinds of visible glass is negative, and the chromatic power is all positive.

### 4.3. Efficient Optimization of Mirror and Housing Material

Table 2 lists the thermal expansion coefficients of common mirror and housing materials for aerial cameras.

Based on Table 2, there are four commonly used mirror materials and four commonly used housing materials, so there are 16 combinations of mirror and housing materials. The optical system includes four lenses, and the number of combinations of equivalent single lenses is $C_{4}^{3}=4$. Therefore, $d_{1}=\frac{d\Delta }{dT}$, the distance $d_{2}$ from $L_{e,origin}(\omega_{e,origin},\gamma_{e,origin})$ to $L_{e}(\omega_{e},\gamma_{e})$ and the distance $d_{3}$ from $L_{e}(\omega_{e},\gamma_{e})$ to $L_{e}^{\prime }(\omega_{e}^{\prime },\gamma_{e}^{\prime })$ can be divided into four groups, with a total of 64 distance ratios.

The lenses are sorted according to the direction of light propagation, namely lens 1 (H-K9L), lens 2 (H-K9L), lens 3 (TF3), and lens 4 (H-K9L). When the single lens is lens 1, $L_{j}\left(\omega_{j},\gamma_{j}\right)$ is (15.58, −8.46), $L_{e}\left(\omega_{e},\gamma_{e}\right)$ is (0.4985, −23.3928), $\Delta =\left(\alpha -\alpha_{m}\right)\times 112.4414$; when the single lens is lens 4, $L_{j}\left(\omega_{j},\gamma_{j}\right)$ is (15.58, −8.46), $L_{e}\left(\omega_{e},\gamma_{e}\right)$ is (0.931601, −22.964), $\Delta =\left(\alpha -\alpha_{m}\right)\times (-318.211)$. It can be seen that when the single lens is lens 1 or lens 4, Δ is larger, and only the distance ratios of SC and INA are meaningful on the athermal map.

Therefore, when the combination of the mirror and housing material is selected by the comprehensive weight, ignore the case that the single lens as lens 1 and 4. When the single lens is lens 2 or lens 3, there are 32 kinds of distance ratios. Taking lens 2 as the single lens as an example, the athermal map of 16 material combinations is shown in Figure 10, where the distance between the red point $L_{1}\left(\omega_{j},\gamma_{j}-\Delta \right)$ and the red * $L_{2}\left(\omega_{j},\gamma_{j}-\Delta -\frac{d\Delta }{dT}\right)$ is $d_{1}$, the distance between the blue point $L_{e,origin}\left(\omega_{e,origin},\gamma_{e,origin}\right)$ and the green point $L_{e}\left(\omega_{e},\gamma_{e}\right)$ is $d_{2}$, and the distance between the green point and the green * $L_{e}^{\prime }(\omega_{e}^{\prime },\gamma_{e}^{\prime })$ is $d_{3}$.

It can be obtained from the definition of the comprehensive weight, and the distance is inversely proportional to the size of the comprehensive weight. The number of rows in Figure 10 represents the choice of mirror material, and the number of columns represents the choice of housing material. It can be found from Figure 10 that when the housing materials are INA and CF, the distribution of points representing distances is more concentrated; from the row point of view, there is no obvious distribution relationship between the choice of mirror material and the degree of point concentration. Therefore, the housing material has a great influence on the athermal potential of the catadioptric optical system.

Using the above calculation method, the comprehensive weights of the single lens as lenses 2 and 3 are added, as shown in Figure 11: The abscissa 1–16 represent the above 16 material combinations, respectively, the ordinate is the comprehensive weight, B is the case where the single lens is lens 2, and C is the case where the single lens is lens 3.

It can be seen from Figure 11 that the range of the weights for 16 material combinations is [2.2, 5.2]. The closer the value is to 5.2, the stronger the athermal ability of the combination of the mirror and housing material; on the contrary, the closer the value is to 2.2, the weaker the athermal ability of the combination of the mirror and housing material. Based on Figure 11, it can be seen that the SC + INA combination has the strongest athermal ability. The combination of mirror and housing material in the initial system is FS + TA, which is the 15th combination in the stacked histogram. There are seven types with weights greater than the initial combination, namely NO.2, NO.4, NO.6, NO.7, NO.8, NO.14, and NO.16. Among them, there are three combinations of housing materials of INA, three combinations of housing materials of CF, and one combination of housing materials of TA. From the perspective of practical engineering applications, the density of INA is 8.1 g/cm^3^, which is 1.8 times that of TA and 2.9 times that of AA. If the housing material is all INA, the weight of the optical system will exceed the design requirements. At the same time, when SC is chosen as the mirror material, it usually increases the cost. Therefore, the best combination of mirror and housing materials is NO.4 (GC + CF) and NO.16 (FS + CF), which can realize that the system not only meets the requirements of engineering applications but also has better athermal ability.

### 4.4. Efficient Optimization of Lens Materials

The weight of the FS + CF combination is 4.65, and the weight of the GC + CF combination is 4.54, so the combination of FS + CF is selected. The athermal map of the combination of mirror and housing material as FS + CF is shown in Figure 12, in which the solid red line is determined by $H\left(0,-\alpha_{m}\right)$ and $L_{e}\left(\omega_{e},\gamma_{e}\right)$, and the red dotted line is determined by $H\left(0,-\alpha_{m}\right)$ and $L_{e}\left(\omega_{e},\gamma_{e}\right)$ after offset by Δ, and the solid green line is determined by $H\left(0,-\alpha_{m}\right)$ and $L_{e}^{\prime }(\omega_{e}^{\prime },\gamma_{e}^{\prime })$. At this time, $d_{1}$, $d_{2}$, and $d_{3}$ are small, and the combination of the mirror and housing material has a strong athermal ability. In order to further improve the temperature adaptability of the system, the lens material should be optimized so that the blue point is adjusted to be near the position of the green point.

It can be seen from No.16 in Figure 11 that the weight of lens 3 (TF3) is larger, so lens 3 is used as a single lens, and lens 1 (H-K9L), lens 2 (H-K9L), and lens 4 (H-K9L) are used as equivalent single lenses for material replacement. When the lens material is the same, the distance from the material to the H-L_e_ line is the same, and the replacement lens cannot be determined. Therefore, it is proposed to select the replacement lens with the ratio of the weighted optical power as the reference index, as shown in Equation (21).
(21)$$M_{i}=\frac{\left|\phi_{i}^{\prime }\right|}{\sum_{1}^{k}\left|\phi_{i}^{\prime }\right|}$$

Both the achromatic and athermal equations are related to the weighted optical power of the lens. When the weighted optical power is large, the lens can effectively adjust the thermal difference distribution of the system with only a small change of material properties.

At this time $M_{1}=0.006911$, $M_{2}=0.990647$, and $M_{4}=0.002442$. The weighted power ratio of lens 2 is much larger than that of lens 1 and lens 4, so lens 2 is the main factor for adjusting the position of the equivalent single lens. The athermal glass optimization map is shown in Figure 13.

It can be seen from Figure 13 that the position of lens 2 on the athermal map is blue +, which coincides with the blue dot. In order to move the blue dot $L_{e,origin}(\omega_{e,origin},\gamma_{e,origin})$ to the green dot $L_{e}\left(\omega_{e},\gamma_{e}\right)$, replace lens 2 with H-ZK1 (15.88, −6.10).

Redistribute optical power after lens material optimization. At this time, $L_{j}\left(\omega_{j},\gamma_{j}\right)$ is (22.68, −1.43), $L_{e,origin}(\omega_{e,origin},\gamma_{e,origin})$ is (15.88, −6.10), $\Delta =\left(\alpha -\alpha_{m}\right)\times (-1.11611)=4.933211$, and $\frac{d\Delta }{dT}=\frac{\left(\gamma_{j}+\alpha \right)\left(\alpha -\alpha_{m}\right)\left(\varphi_{1}^{\prime }+\varphi_{2}^{\prime }\right)}{\phi_{j}^{\prime }}=-4.19323$. The final athermal map is shown in Figure 14.

It can be seen from Figure 14 that the blue dot and the green dot basically coincide.

### 4.5. Design Results and Analysis

Table 3 shows the optical properties of the components in the final athermal optical system.

From the comparison between Table 1 and Table 3, it can be seen that the material of element 4 has changed, the chromatic power increases, the thermal power increases, and the change in optical power is not large.

According to the characteristics of the focal depth, when the defocus of the optical system is less than one time of the focal depth, the imaging quality of the optical system basically does not change. For an optical system with a center wavelength of 586 nm and an F-number of 4, the focal depth is 0.01875 mm. When the ambient temperature changes, the thermal defocus is the difference between the back focal length BFL and the flange distance FBD. The thermal defocus curves before and after athermalization are shown in Figure 15.

It can be seen from Figure 15 that the thermal defocus of the initial system at most temperatures is much larger than the focal depth. After the athermal design, the thermal stability of the system is significantly improved. It can be seen from Figure 15 that the thermal defocus of the athermal system in the temperature range of −40–60 °C is less than 0.004 mm, and all of them are within the focal depth.

The MTF performance of the final athermal optical system is shown in Figure 16.

It can be seen from Figure 16 that at 20 °C, −40 °C, and 60 °C, the MTFs of all fields of view in the final athermal optical system are greater than 0.39 at 68 lp/mm. In this paper, through the optimization of the combined material (the mirror and housing) and the optimization of the lens material on the athermal map, the thermal defocus of the final optical system is much less than the focal depth, and the MTFs of the optical system basically does not change with temperature, which achieves a suitable athermal effect.

## 5. Discussion

In order to clearly prove that the method in this paper can achieve material optimization more efficiently, the athermal design is carried out in the optical software to evaluate the effect of material optimization on the transfer function of the system at different temperatures. The refractive indices of homologous materials are relatively close, and the Abbe number does not change much. Therefore, when optical software is used for athermal design, homologous materials are usually selected for material optimization, which can change the temperature of the system while ensuring that the system parameters do not change much. Multiple structures with different temperatures are established, and the curvature radius, thickness, and interval between the lenses are set as variables, and automatic optimization is carried out in the optical design software. The left side of the initial athermal equation is −0.021, and the right side is −0.014. In order to balance the influence of temperature on the optical system, the lens material with a large thermal power should be used as the optimized material. The glass material of lens 2 is H-K9L, and the homologous materials with larger thermal power are K8 (17.6, −7.95), H-K5 (15.78, −7.71), and H-K3 (15.44, −3.69) in order. The MTF of the optical system after the third material optimization is shown in Table 4.

It can be seen from Table 4 that the MTF (at 68 lp/mm) at −40 °C of the initial system is low, and when lens 2 is H-K8, the MTFs of the system are optimized at both −40 °C and 60 °C. As the thermal power continues to increase, the results of multiple structural optimizations at this time do not continue to increase. In the third optimization, the MTFs decreased significantly at high and low temperatures, which was due to the large change in the thermal power of homologous glass materials, which changed from −7.71 to −3.69. Therefore, in the case of athermal design, the temperature adaptability of the system can be improved a little through the optimization of the homologous materials. Through comparison, we can find that the method in this paper achieves a better athermal effect by only one material optimization.

## 6. Conclusions

Aiming at the problem that the existing methods are difficult to achieve the athermal design of the catadioptric optical system, this paper proposes an athermal design method based on the efficient optimization of materials. The material optimization of the catadioptric optical system is carried out in the athermal map, and the expression of the thermal power offset of a single lens is derived, and it is pointed out that the thermal power offset of a single lens is the key factor affecting the achromatic and athermal design of the catadioptric optical system.

In this paper, the optimization of the mirror and housing material is carried out through the comprehensive weight of the offset change with temperature and the position change of the equivalent single lens and the problem of how to match the mechanical material in the athermalization of the catadioptric optical system is solved. This weight can be used as the evaluation index of athermal design, which points out the direction for the optimization of mechanical materials in catadioptric optical systems. In addition, the selection of lens materials is carried out by the ratio of the weighted optical power, and the efficient optimization of lens materials is realized by making the initial equivalent single lens position coincide with the ideal equivalent single lens position in the athermal map, which avoids the situation that the athermal efficiency is low when the thermal power changes greatly, and improves the athermal efficiency.

Through a series of models and formulas established in this paper, it can be concluded that the offset of the thermal power of a single lens is related to the weighted optical power of the single lens, the primary and secondary mirrors, and the thermal expansion coefficients of the mirror and housing materials. Different from the athermal design of the previous catadioptric system, the method proposed in this paper considers the influence of the combination of the mirror and the housing material, which breaks the limitation of the athermal design of the catadioptric system. At the same time, the linear relationship in the athermal map of the refractive system is extended to the catadioptric optical system so that the applicable scope of the graphic method is wider. In summary, the achromatic and athermal design method proposed in this paper makes the system have better temperature adaptability by realizing efficient optimization of materials, which has certain theoretical and practical value.

## Figures and Tables

**Figure 1 sensors-23-01754-f001:**
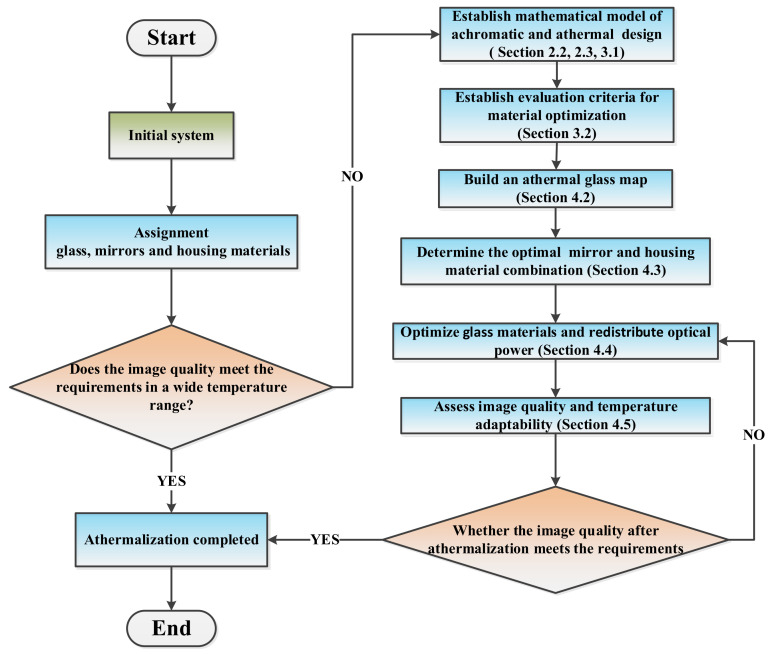
Flow chart of athermal design for catadioptric optical systems.

**Figure 2 sensors-23-01754-f002:**
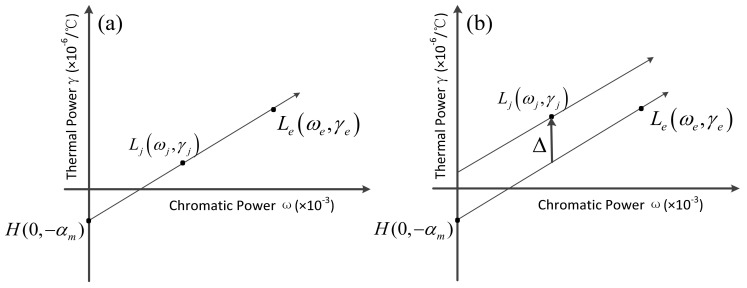
Achromatic and athermal conditions on an athermal map of (**a**) refractive optical system and (**b**) catadioptric optical system.

**Figure 3 sensors-23-01754-f003:**
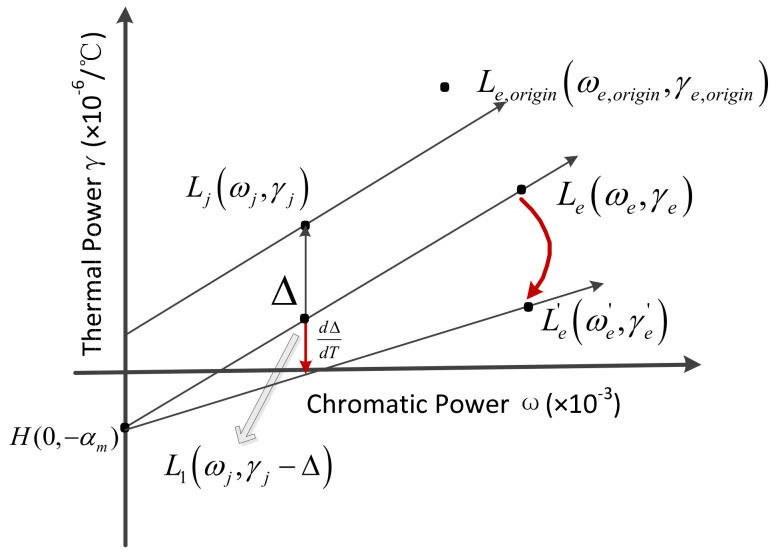
Achromatic and athermal conditions on an athermal map when the offset changes with temperature.

**Figure 4 sensors-23-01754-f004:**
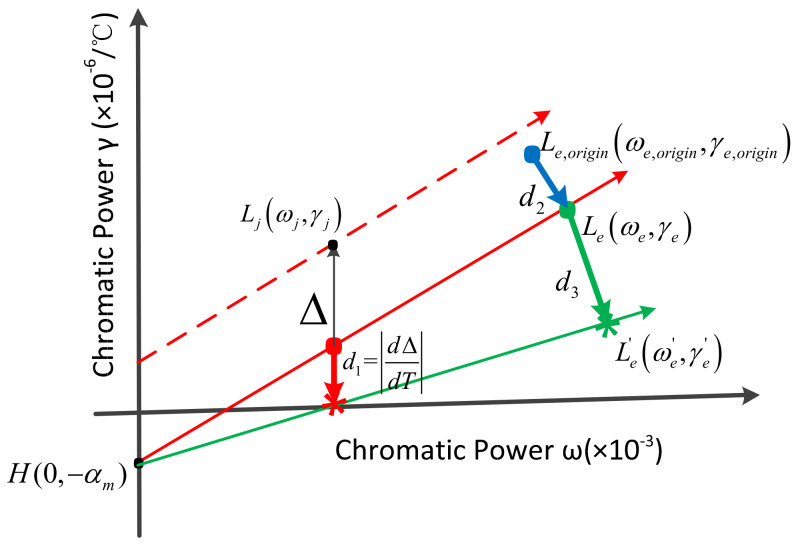
Evaluation method of mirror and housing material combinations.

**Figure 5 sensors-23-01754-f005:**
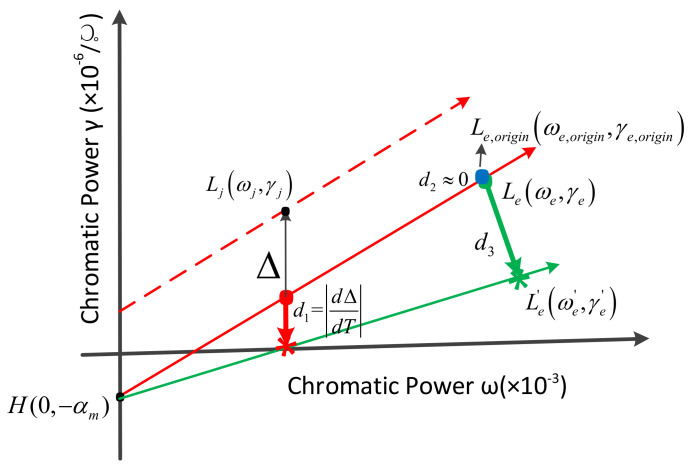
Evaluation method of lens materials.

**Figure 6 sensors-23-01754-f006:**
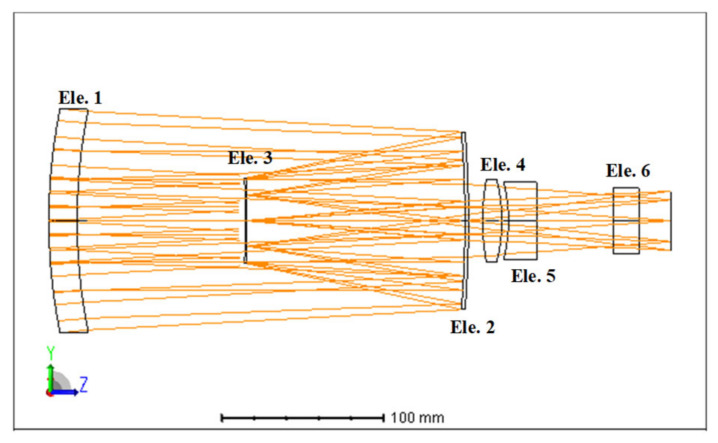
Layout of the initial optical system.

**Figure 7 sensors-23-01754-f007:**
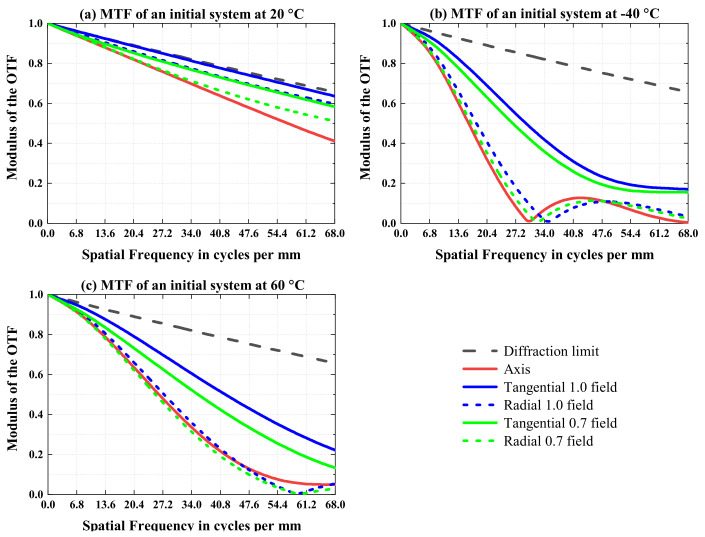
MTF performance of the initial optical system at temperatures of (**a**) 20 °C, (**b**) −40 °C, and (**c**) 60 °C.

**Figure 8 sensors-23-01754-f008:**
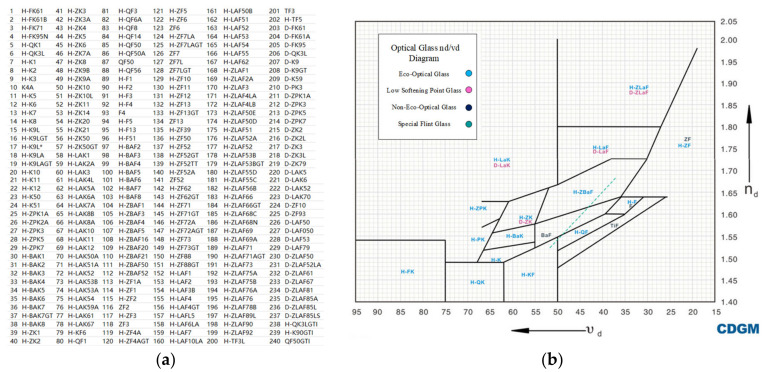
Chengdu Guangming glass (**a**) material catalog, (**b**) n_d_/v_d_ map.

**Figure 9 sensors-23-01754-f009:**
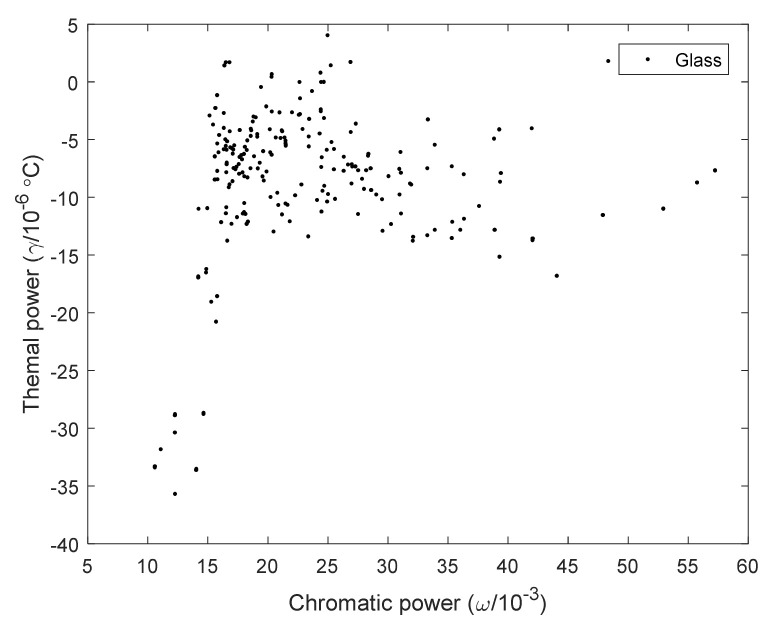
Athermal glass map of the Chengdu Guangming catalog calculated by equations.

**Figure 10 sensors-23-01754-f010:**
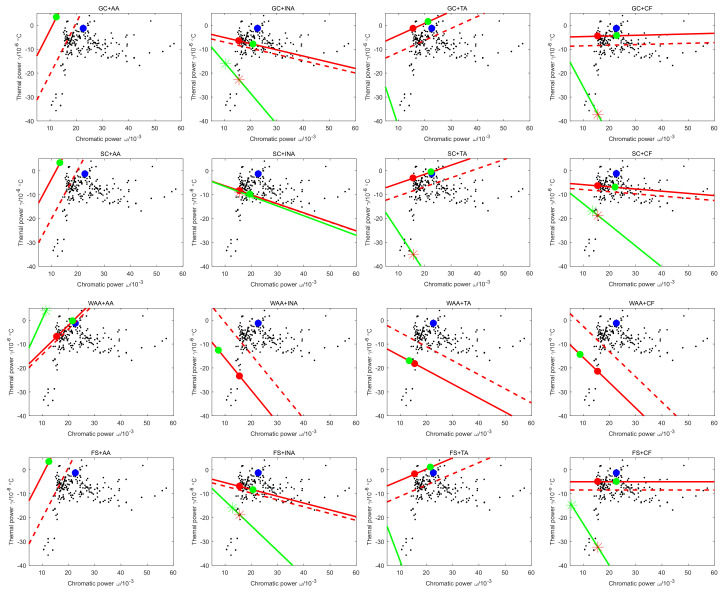
Athermal glass map of different material combinations.

**Figure 11 sensors-23-01754-f011:**
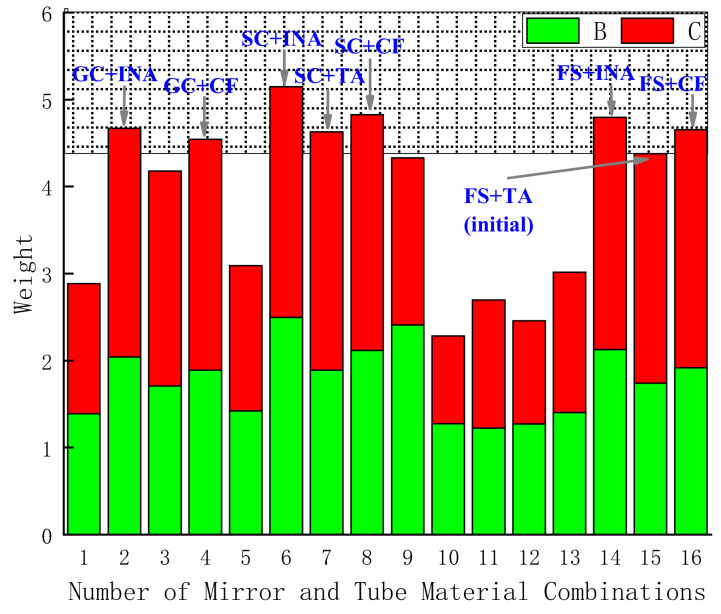
Weight map for different material combinations.

**Figure 12 sensors-23-01754-f012:**
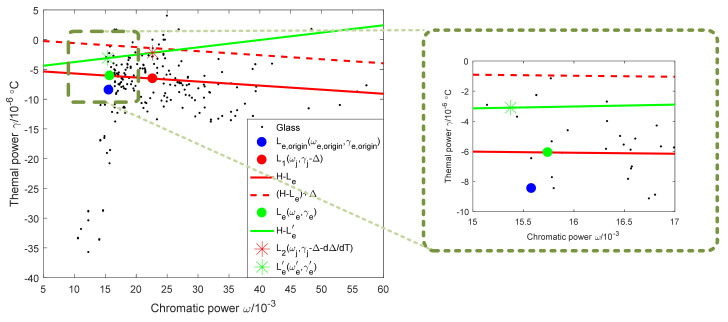
Athermal map of the combination of mirror and housing material as FS + CF.

**Figure 13 sensors-23-01754-f013:**
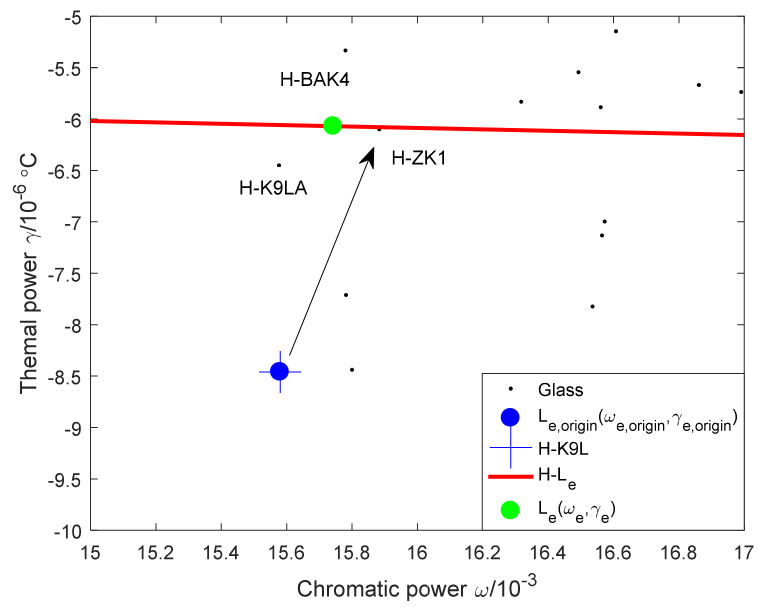
Athermal glass optimization map.

**Figure 14 sensors-23-01754-f014:**
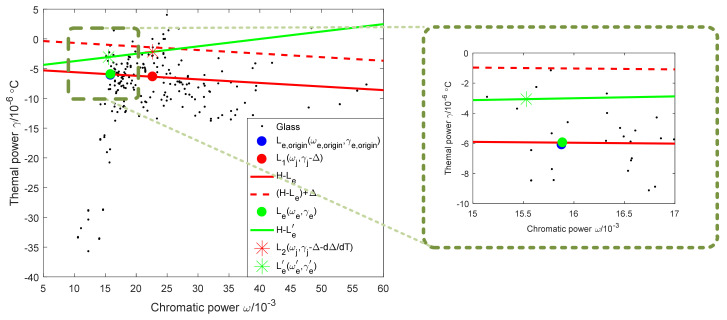
Final athermal map.

**Figure 15 sensors-23-01754-f015:**
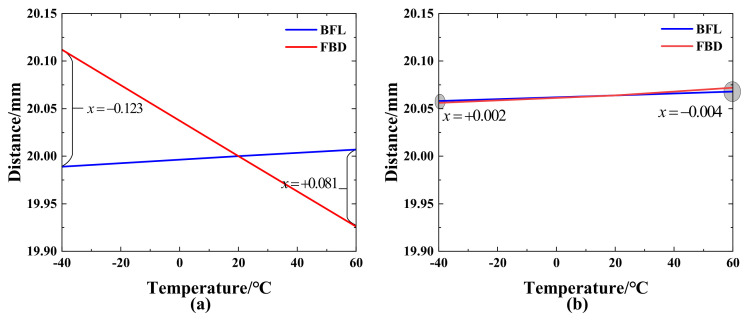
Thermal defocus curves (**a**) initial system (**b**) athermal system.

**Figure 16 sensors-23-01754-f016:**
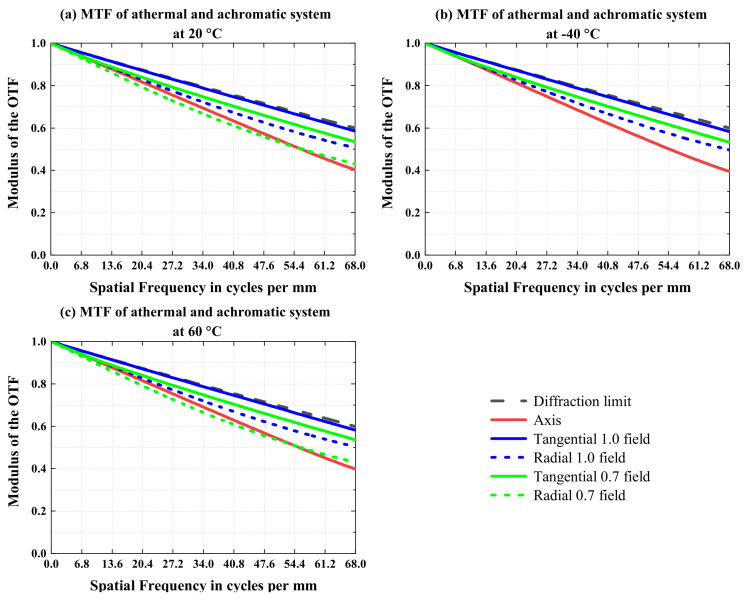
MTF performance of the final athermal optical system at temperatures of (**a**) 20 °C, (**b**) −40 °C, and (**c**) 60 °C.

**Table 1 sensors-23-01754-t001:** Optical properties of elements in the initial optical system.

Element	Material	ω $/(\times 10^{-3})$	γ $/(\times 10^{-6}/{}^{\circ}C)$	ϕ (mm^−1^)	h/mm
1	H-K9L	15.58	−8.46	0.000018	43.7499
2	MIRROR	0	−0.58	0.002858	42.7784
3	MIRROR	0	−0.58	−0.001307	25.7322
4	H-K9L	15.58	−8.46	0.008754	12.8951
5	TF3	22.68	−1.43	−0.006848	11.1802
6	H-K9L	15.58	−8.46	−0.000075	3.71021

**Table 2 sensors-23-01754-t002:** Thermal expansion coefficients of common mirror and housing materials.

Mirror Material	$\alpha /\left(10^{-6}/{}^{\circ}C\right)$	Housing Material	$\alpha /\left(10^{-6}/{}^{\circ}C\right)$
Glass-ceramic (GC)	0.05	Aluminum alloy (AA)	23.6
Silicon carbide (SC)	2.4	Iron nickel alloy (INA)	2.5
Wrought aluminum alloy (WAA)	21.6	Titanium alloy (TA)	9.1
Fused silica (FS)	0.58	Composite fiber (CF)	5

**Table 3 sensors-23-01754-t003:** Optical properties of the components in the final athermal optical system.

Element	Material	ω $/(\times 10^{-3})$	γ $/(\times 10^{-6}/{}^{\circ}C)$	ϕ (mm^−1^)	h/mm
1	H-K9L	15.58	−8.46	0.000020	43.7500
2	MIRROR	0	−0.58	0.002803	42.4249
3	MIRROR	0	−0.58	−0.001199	25.8701
4	H-ZK1	15.88	−6.10	0.009055	13.1561
5	TF3	22.68	−1.43	−0.006485	12.1529
6	H-K9L	15.58	−8.46	−0.000215	3.77475

**Table 4 sensors-23-01754-t004:** The MTF at 68 lp/mm of the optical system after multiple structure optimization.

Number of Optimizations	MTF at 20 °C	MTF at −40 °C	MTF at 60 °C
Initial	0.41	0.21	0.37
1(H-K8)	0.41	0.25	0.39
2(H-K5)	0.41	0.24	0.32
3(H-K3)	0.41	0.15	0.20

## Data Availability

Not applicable.

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
