# Peer review of "Achromatic and Athermal Design of Aerial Catadioptric Optical Systems by Efficient Optimization of Materials"

_sensors, 2023, doi:10.3390/s23041754_

Round 1

Reviewer 1 Report

I cannot recommend the article for publication in the present state. Please find my comments below.

MAJOR:

-          The article has quite large theoretical part outlining the «achromatic and athermal theory». Surprisingly, the section doesn’t contain any reference to works by other researchers. For example, the equations (1) can be found in the article http://dx.doi.org/10.1364/OE.24.018049 . I recommend to expand the reference list and provide more references to other works taking into account that some theory has already been published elsewhere. This comment is actual to other sections of the article too.

-          In the article, I didn’t find much information on sensors. Subsequently, I don’t understand why the article was submitted to «Sensors». In my opinion, the article more fits the scope of «Optics». Please provide more information on specific use of «aerial catadioptric optical systems» in sensing applications.

MINOR:

-          The abstract is definitely too long. I’d recommend to follow the journal’s guide and shorten it down to 200 words. Please specify which materials were the best for the system, according to your design.

-          In table 1, I didn’t understand which elements the numbers from 1 to 6 stand for. Please assign them in Fig. 6.

-          In the section 4.2, the athermal glass map is shown in Fig. 8. I found no information on chemical compositions of the glasses. However, this information is essential.

TYPOS and STYLE:

-          Page 6 line 203: missing «space» after «Δ»

-          Please combine Table 2 and 3 in single one.

-          Page 17 line 467: «Discussion»

-          Page 18 lines 485 and 486: check spaces before «o

Reviewer 2 Report

See enclosed file

Round 2

Reviewer 1 Report

Dear Authors,

Thank you very much for your answers. You addressed all my questions. I recommend your article for publication.

Author Response

Thank you for your comment. We have revised the English as follows:

line 130: “housing” is changed to “housing”.

line 144: The sentence is changed to “Therefore, how to efficiently optimize the mirror and housing materials according to the actual situation is an important research topic for people designing athermal catadioptric optical system.This is an important topic only for people designing athermal catadioptric optical system and not necessarily the reader !”

line 157: The sentence is changed to “The above athermal method based on the athermal map is conducive to improve the efficiency of lens material optimization.”

line 173-196: The sentence is changed to “In order to select the optimal mirror and housing material more quickly, a combined analysis of the mirror and housing material is proposed. The athermal ability of the combined material is evaluated based on the comprehensive weight of the change of the thermal power offset of the single lens with temperature and change in position of the equivalent single lens.”

Reviewer 2 Report

Dear Authors,

Thank you for the revised article. When reading it again, I felt that figure 8 would be nicely replaced by an Abbe diagram with the name placed on the correct position, because a list of names doesn't help.

Otherwise, in my opinion, the article is too long and in a next submission, please use the possibility to add annexes!

Best regards

Author Response

(1)Thank you for your comment. We have added the nd/vd map of Chengdu Guangming Glass in figure 8(b).

(2)Thank you for your comment. I don't agree with your comment of adding annexes as much as possible. I think the paper has added sufficient references and has a complete structure. Do you have any more specific suggestions?